# An improved North-South synchronization of ice core records around the 41 kyr beryllium-10 peak

Grant M. Raisbeck[1], Alexandre Cauquoin[2], Jean Jouzel[2], Amaelle Landais[2], Jean-Robert Petit[3], Vladimir Y. Lipenkov[4], Juerg Beer[5], Hans-Arno Synal[6], Hans Oerter[7], Sigfus J. Johnsen[8, #], Jorgen P. Steffensen[8], Anders Svensson[8], Françoise Yiou[1]

[1]CSNSM, CNRS, Université Paris-Saclay, Bats 104-108, 91405 Campus, Orsay, France
[2]LSCE/IPSL, CEA-CNRS-UVSQ), CEA Saclay Orme des Merisiers, 91191 Gif-sur-Yvette, France
[3]LGGE, CNRS, BP 96, 38402, St Martin d'Hères cedex, France
[4]Arctic and Antarctic Research Institute, 38 Bering St., St. Petersburg 199397, Russia
[5]Eawag, Überlandstrasse 133, Postfach 611, 8600 Dübendorf, Switzerland
[6]Laboratory of Ion Beam Physics, ETH Zurich, 8093 Zurich, Switzerland
[7]Alfred Wegener Institute for Polar and Marine Research, 27570 Bremerhaven, Germany
[8]Centre for Ice and Climate, Niels Bohr Institute, University of Copenhagen, Copenhagen, Denmark
[#]deceased

*Correspondence to*: Grant M. Raisbeck (Grant.Raisbeck@csnsm.in2p3.fr)

**Abstract.** Using new high resolution $^{10}$Be measurements in the NGRIP, EDML and Vostok ice cores, together with previously published data from EDC, we present an improved synchronization between Greenland and Antarctic ice cores during the Laschamp geomagnetic excursion ~41 kyr ago. We estimate the precision of this synchronization to be ± 20 years, an order of magnitude better than previous work. We discuss the implications of this new synchronization for making improved estimates of the depth difference between ice and enclosed gas of the same age (Δdepth), difference between age of ice and enclosed gas at the same depth (Δage) in the EDC and EDML ice cores, spectral properties of the $^{10}$Be profiles and phasing between Dansgaard/Oeschger-10 (in NGRIP) and AIM-10 (in EDML and EDC).

## 1 Introduction

In a previous study, Raisbeck et al. (2007) have synchronized the Greenland GRIP and Antarctic EPICA Dome C (EDC) ice cores using the structured peak of cosmogenic $^{10}$Be resulting from a combination of the low geomagnetic intensity and variable solar activity during the Laschamp geomagnetic excursion ~41 kyr ago (Lascu et al., 2016 and references therein). The estimated precision of this synchronization was 200 years, due mainly to uncertainties associated with the GRIP $^{10}$Be record (sample time resolution of 25-50 years, corrections for loss of $^{10}$Be on 0.4 micron filters for some samples, several missing samples). We improve this situation significantly here by using a much higher resolution (5-10 year) $^{10}$Be profile in the NGRIP ice core. In addition we report and synchronize high resolution $^{10}$Be profiles from the peak region of two other Antarctic cores, EPICA Dronning Maud Land (EDML) and Vostok. Maps showing the locations and a brief description of drilling operations for the ice cores studied here can be found in Jouzel (2013). With these improvements, we estimate the uncertainty on the synchronization to be ± 20 years at our new tie points, and ± 35 years over the whole Laschamp event. This precision is supported by the study of Svensson et al. (2013) who, on the basis of

our earlier synchronization, have identified 3 volcanic signals which they believe are common to the Greenland and Antarctic cores.

## 2 Methods

A continuous series of 55 cm long bag samples from the NGRIP core were available from 2102 m to 2140 m. Each of these were cut into five 11 cm samples weighing ~100 g, and representing 5-10 years. The samples were treated and measured on the Gif sur Yvette Tandetron AMS facility as described in Raisbeck et al. (2007). AMS measurements were made relative to NIST $^{10}$Be/$^9$Be standard SRM 4320, assuming a nominal value of $2.68 \times 10^{-11}$. While the value of $2.79 \times 10^{-11}$ has now been adopted by many groups for this standard, for consistency we continue to use the value that was used for all the previous measurements at the Gif sur Yvette Tandetron. Since this factor is constant, it has no effect on our synchronization procedures. An average of 400 $^{10}$Be ions were counted for each sample, and the estimated precision (not including the uncertainty of the standard) was ~7%.

Vostok was one of the sites (the other being the original Dome C site) where the $^{10}$Be peak discussed here was initially found (Raisbeck et al., 1987). Although the estimated date was younger than presently accepted, and the origin uncertain, it was already pointed out there that such a peak was a potentially interesting stratigraphic marker. Those initial measurements were made on discontinuous samples from the Vostok 3G core. We subsequently made measurements on a nearly continuous sequence of 1 m samples from the 4G core, which showed a double humped peak (Raisbeck et al., 1992). It was in an effort to look for even finer structure that the measurements shown here were made. These samples were ~10 cm in length and weighing >500g, were taken from the 5G core between 580-620 m. As can be seen in Fig. 1d, the sampling depth is not centered on the peak. This is because the sampling was based on the initial identification in 3G and, as we discovered, there is apparently a ~10 m offset between 3G and 5G. Since these measurements were carried out >15 years ago, also at the Gif AMS facility, the sample preparation and measurement procedures are those described in Raisbeck et al. (1987). For almost all samples at least 1000 $^{10}$Be ions were counted, leading to an estimated measurement uncertainty of ~ 6%.

The EDML $^{10}$Be measurements were performed at ETH Zurich with a sample preparation that is based on the common procedure already applied for the GRIP ice core (Yiou et al., 1997; Muscheler et al., 2004). Samples of 25 cm length corresponding to ~15 year resolution and a typical weight of 110 g were melted and processed without any further filtering. The $^{10}$Be/$^9$Be ratios were measured using the compact AMS facility Tandy at ETH Zurich (Müller et al., 2010) resulting in uncertainties <3%. Results were normalized to the internal standard S2007N with a reference value of $^{10}$Be/$^9$Be $= (28.1 \pm 0.8) \times 10^{-12}$ (Christl, et al., 2013). The EDC $^{10}$Be measurements have been described in detail previously (Raisbeck et al., 2007).

The climate records are provided by high resolution isotopic profiles (either $\delta$D or $\delta^{18}$O) measured along these ice cores. $\delta$D is used as primary climate indicator for Vostok and Dome C while $\delta^{18}$O is used for NGRIP and EDML. In addition to existing water isotopic records, new high resolution measurements have been performed for this study on the Vostok 5G ice core (10 cm) and EDML (50 cm). These isotopic data are shown in Fig. 1a to 1d, with in addition a smooth curve (see legend).

From the [10]Be concentrations, [10]Be fluxes of the 4 cores were calculated using accumulation rate estimates from water isotopes, ice thinning and dating constraints by Parrenin et al (2007) for EDC, Ruth et al. (2007) for EDML, Parrenin et al. (2004) for Vostok, and deduced from layer counting (Svennson et al. 2008) combined with the ss06bm thinning model of Johnsen et al., (2001) for NGRIP. Alternative accumulation rate determinations were obtained more recently through dating exercises such as for the AICC2012 timescale. These determinations do not show significant differences with previous determinations over our period of interest (Bazin et al., 2013; Veres et al., 2013; Andersen et al., 2006; Lemieux-Dudon et al., 2015). Only for the NGRIP ice core was it recently suggested that the mean accumulation rate of the MIS 3 could be overestimated by 20-30% (Guillevic et al., 2013; Kindler et al., 2014; Lemieux-Dudon et al., 2015). Still, even when using this modified accumulation rate, the shape of the variations of [10]Be flux on the Laschamp event recorded at NGRIP is not significantly affected. This is because for our purpose, it is the relative accumulation rates which are important.

## 3 Results

The results for the [10]Be concentrations (see Supplementary Material) and fluxes of the 4 cores are shown in Fig. 1 as a function of their depth along with their stable isotope profiles. As previously noted (Yiou et al. 1997, Raisbeck et al 2007) all cores show a highly structured peak of [10]Be centered on AIM-10 (EDC, EDML, Vostok) or DO-10 (NGRIP). It is this structure that allows us to make multiple synchronizations over the whole width of the peak, as shown in Fig. 2. This synchronization was done in two ways. Initially we selected 5 tie points (4 peaks and one valley, Table 1 and Fig. 2) which appeared to us to be clearly common to all the profiles, and synchronized using the AnalySeries program of Paillard et al (1996). Subsequently, in an effort to avoid the subjectivity of choosing tie points, we did the synchronization using the MATCH protocol of Lisiecki and Lisiecki (2002). The difference between these two procedures never differed by more than 42 years for NGRIP-EDC and 27 years for NGRIP-EDML. The maximum difference for the two procedures was for the period between tie points C and D, and was very small (<10 years) near the tie points. This confirms our visual impression that the chosen tie points are indeed the most robust common features of the profiles. For Vostok, we were unable to get a satisfactory synchronization for the MATCH protocol without imposing severe restrictions. This is probably due to several missing sample sections for that record.

While we believe there is considerable evidence that flux is a better measure of [10]Be (and other trace species) deposition in polar cores, at the suggestion of a reviewer we have repeated the Match synchronizations of EDC and EDML with NGRIP using concentrations. We found that the 4 peaks and valley corresponding to the tie points shown in Fig 2 did not vary by more than 20 years compared to those found using fluxes. In summary we believe our [10]Be synchronization is quite robust to the choice of raw or smoothed accumulations, or even the use of concentration.

In order to estimate the precision of our synchronization, we have used two procedures. Initially we adopted a procedure which involved looking at the correlation of EDC with EDML based on their [10]Be synchronization with NGRIP, and compared it with the direct correlation of EDC and EDML by Severi et al. (2007) using common volcanic peaks. The results are shown in Fig 3a. As can be seen the agreement is excellent. For the 6

volcanic peaks of Severi et al. (2007) that fall within our [10]Be tie points, the average difference between the observed and predicted depths corresponds to 4.7±6.2 years (2σ standard deviation). One of the reviewers (C. Buizert) has suggested that this "would provide a true test of the uncertainty in the [10]Be synchronization". However we feel that, while it does strongly support our synchronization procedure, it may underestimate the uncertainty between NGRIP and the Antarctic cores, as discussed below.

As a second estimate of the precision of our synchronization, we can look at the volcanic correlations of Svensson et al. (2013). Starting from an earlier [10]Be synchronization, and layer counting results in both NGRIP and EDML, those authors identified 3 volcanic spikes they believed were common to NGRIP-EDML, 2 of which

are linked to EDC by Severi et al. (2007). In Table 1 and Fig. 3b,c, we show the position of those events as found by our improved [10]Be synchronization. For EDML the differences are 66 cm (~44 yr) for L1, 51 cm (~28 yr) for L2 and 44 cm (~24 yr) for L3. In EDC the differences compared to those given by Svensson et al. (2013) are 27 cm (~23 yr) for L2 and 36 cm (~31 yr) for L3. As mentioned above, we believe our tie points have the most robust synchronized ages. It is therefore reasonable to expect the difference in predicted ages to increase

with distance from those tie points. However L1 is an exception, having the largest difference (66 cm) despite being the nearest (~1 m) to a tie point (BeC). If that tie point is correct, it is thus reasonable to question whether L1 is indeed a bipolar event. In fact L1 was not listed as one of those used by Severi et al. (2007) to correlate EDC with EDML.

For the other two volcanic peaks, it can be noted that the age differences are systematic, having an average value of 27±7 (2σ) years for L2 and L3. Thus the relative predicted ages between EDML and EDC for L2 and L3 differ by less than 10 years, consistent with their identification as the same event by Severi et al. (2007). However, 27±7 years is significantly larger than the 4.7±6.2 year precision found above. We must therefore conclude that either L2 and L3 of Svensson et al. (2013) are also not bipolar, or that 4.7±6.2 years

underestimates our real uncertainty between NGRIP and EDC/EDML. We tend to favour the second explanation. The reason is that the [10]Be peaks involved are irregular and have durations of the order of 100 years. Given this, and the [10]Be sampling resolution, it seems to us unlikely that they can be synchronized to a precision of less than 10 years. Our synchronization procedure, either by eye (AnalySeries) or Match protocol, will maintain a tight correlation between EDC and EDML when synchronizing with any feature in NGRIP. However,

if for some reason (higher and more variable ice accumulation rate, higher sampling resolution), the form of the [10]Be flux peaks at NGRIP differ slightly from those in EDC/EDML, then the correlation between NGRIP and the Antarctic cores may be offset by several decades. For this reason, we prefer to be conservative and adopt as our uncertainty the value implied assuming that the volcanic peaks L2 and L3 of Svensson et al. (2013) are indeed bipolar.


In summary, if L2 and L3 are indeed bipolar events, and L1 is not, we estimate from the above that the uncertainty in our synchronization is ≤35 years over the whole Laschamp event, and ≤20 years at our tie points. This is an order of magnitude better than in (Raisbeck et al. 2007). It can also be compared with the 2 sigma uncertainty in the methane interpolar synchronization (±73 years) estimated by Buizert et al. (2015) at this age.

This shows that, at least in regions where there is significant structure in [10]Be profiles, it is possible to link

Greenland and Antarctic climate records with decadal precision. As pointed out by Svensson et al. (2013), using such correlation as a framework, it may be possible to find other common volcanic signals in ice cores from the two hemispheres.

## 5   4 Spectral properties of the [10]Be profiles

As indicated in the Introduction, the centennial variations in the [10]Be profiles are believed to be caused by variations in solar activity. For example, during the Holocene, Steinhilber et al (2012) found periodicities of 88 years (*Gleissberg* cycle) and 210 years (*de Vries* cycle) in a composite of tree ring [14]C and ice core [10]Be records. For the Laschamp period, Wagner et al. (2001) reported a ~205 year periodicity of [10]Be in the GRIP core. It is thus interesting to do a spectral analysis on the higher resolution and better quality NGRIP 10Be record reported here. This is shown in Fig. 4, where a very significant (> 99%) 200 year signal is found in the Fourier spectrum (REDFIT program of Schulz and Mudelsee (2002)). As can be seen in the wavelet spectra however (MATLAB package of Grinsted et al. (2004)), this periodicity is really only significant for a short interval near the end of the time period studied. While there is also a short interval where an 88 year periodicity appears, this is not significant in the Fourier spectrum.

Since we now have the EDC and EDML [10]Be profiles synchronized on the NGRIP time scale, we might expect these to show similar periodicities. Somewhat to our surprise, however, this is only partially the case. While both show a ~200 year signal in the same time region of the wavelet analyses, these are barely significant (~90%) in the Fourier spectra. This is because that peak appears to be distorted by longer signals of ~290 years and 350 years in EDC and EDML respectively. This is consistent with the observation of Cauquoin et al. (2014) who found that the 210 year periodicity in the EDML Holocene data used by Steinhilber et al. (2012), the only Antarctic core in their composite, was only significant over short and sporadic time periods. It is also consistent with the fact that Cauquoin et al. (2014) did not find any evidence for this periodicity in [10]Be from EDC during the interval 336-325 ka BP. It thus appears that there may be meteorological effects in the Antarctic (smaller ice accumulation rate?) which are relatively more important than in Greenland, and which tend to diminish or interfere with the [10]Be production signal recorded in the ice, compared to Greenland.

## 5 Implications for ΔDepth and ΔAge Estimates

Using preliminary values of our previous [10]Be synchronization, together with methane correlations, Loulergue et al. (2007) estimated the difference in depth (Δdepth) and age (Δage), between the ice and gas records in EDC and EDML. The discrepancy they found at EDC compared with the classical firn densification model (Goujon et al. 2003) was in fact the first hard evidence that this model was inadequate for glacial ice from low accumulation sites in Antarctica. This had important implications for the EDC4 gas age construction. Loulergue et al. (2007) hence proposed that the Δage obtained by the Goujon et al. (2003) model using temperature and accumulation rate deduced from water isotopes (scenario 1 in Fig. 5 and Loulergue et al., 2007) was not appropriate. Instead, Δage for the EDC gas age was calculated with the Goujon et al. (2003) model but with an artificially higher accumulation rate than the one used in the construction of the EDC3 ice timescale (scenario 4 in Fig. 5 and

Loulergue et al., 2007), so that the calculated Δage at 41 kyr was in agreement with the Δage determination using $^{10}$Be and $CH_4$ synchronization.

In the construction of the AICC2012 chronology (Bazin et al., 2013; Veres et al., 2013), a different approach for the Δage calculation has been chosen. The construction of AICC2012 relies on the use of a bayesian tool, DATICE [Lemieux-Dudon et al., 2010], to optimize the chronology of 5 ice cores using absolute dating constraints and stratigraphic links in the gas and ice phases. The AICC2012 Δage calculation for EDC is hence constrained by the $CH_4$ and $^{10}$Be data as in Loulergue et al. (2007). It also benefits from the recent finding of Parrenin et al. (2012) showing that the firn lock-in depth at Dome C during the last deglaciation was best determined from $\delta^{15}N$ measurements in air trapped in ice cores rather than by outputs of firn modeling. As a consequence, the background scenario used for the lock-in depth and hence the Δage in the construction of the AICC2012 chronology for EDC was based on $\delta^{15}N$ measurements performed at Dome C (Dreyfus et al., 2010; Landais et al., 2013) leading to a significantly smaller Δage than the firn model output for EDC during the glacial period. It should be emphasized that no firn model has been used for the background scenario of the EDC lock-in depth in the construction of AICC2012 which makes the comparison of Δage and Δdepth between the AICC2012 and EDC3 gas timescales not straightforward.

Using the 5 tie points from the present synchronization, we have used a slightly modified version of the technique described by Loulergue et al. (2007) to calculate Δdepth of EDC and EDML. Instead of correlating the methane profiles at a single point, we have correlated the whole profiles using the Match protocol (Fig. 6). Then, a Δage has been determined for each $^{10}$Be peak. The resulting Δages and Δdepths (Fig. 5) are compared to the Δdepth and Δage provided by Loulergue et al. (2007) for the construction of the EDC3 gas timescale. Our results are in reasonably good agreement with the Δage and Δdepths calculated using scenario 4 in Loulergue et al. (2007). Our results are also in good agreement with the AICC2012 Δage at EDC hence validating the chosen approach for delta age calculation. It is clear from Fig 6 that the resolution of the methane data in EDC and EDML, particularly in the period 41.3 – 42.5 kyr, is the limiting factor in calculating the Δage uncertainty.

**6 Implications for Bipolar Seesaw and Stable Isotope Interpretation**

The last glacial period is characterized by a succession of millennial-scale climatic events identified both in Greenland and in Antarctica. In Greenland, the climate shifts are very abrupt with a temperature shift between a cold Greenland stadial (GS) to a warm Greenland interstadial (GIS) designated as Dansgaard-Oeschger events (DO). The DO succession and amplitude seems quite unpredictable even if it has already been suggested that the sequence of the abrupt variability of the last glacial period (length and frequency of GS and GIS) can be affected by the long term (orbital) climatic variability (Schulz et al., 2002; Capron et al., 2010). In Antarctica the temperature changes of the corresponding climatic events are much smoother, and are designated as Antarctic Isotopic Maxima (AIM). A clear relationship has been demonstrated between DO and AIM events: each DO event is associated with an AIM (EPICA community members, 2006; Jouzel et al., 2007) and the temperature/isotopic maximum of the AIM occurs approximately synchronously with the Greenland abrupt temperature increase (Fig 7). The observed relationship between Greenland and Antarctica temperature evolutions led to the proposed theory of the bipolar seesaw (Broecker et al., 1998; Stocker et al., 1998). The

global picture of the bipolar seesaw can be explained by a simple modeling of a slow thermal response of Antarctic temperature to Greenland abrupt warmings and coolings through a heat reservoir in the southern ocean (Stocker and Johnsen, 2003).

The knowledge of the exact relative timing between DO events in Greenland and AIM events in Antarctica is limited by the ice core chronologies. Usually, Greenland and Antarctic ice cores are synchronized using the global atmospheric signals in the air trapped in ice core ($CH_4$, $\delta^{18}O$ of $O_2$). However, the climatic signals are recorded in the ice phase through water isotopic composition. Because the entrapped air is systematically younger than the entrapping ice, by centuries to millennia, accurately determining this age difference is crucial
for the Greenland vs. Antarctica phasing.

Recent studies have permitted to significantly decrease the age uncertainty on Greenland vs. Antarctica climatic sequences. This has been done through an increase in the number of stratigraphic links between Greenland and Antarctic ice cores (e.g. Schüpbach et al., 2011; Svensson et al., 2013) and new high resolution Antarctic records
at relatively high accumulation sites, hence with small ice – air age differences (WAIS Divide Project Members 2015). These recent improvements have revealed a refined sequence of Greenland vs. lower latitudes and Antarctic climatic changes during the last glacial period. First, for GS associated with an occurrence of Heinrich events, it has been evidenced that a decoupling exists between the Greenland or high latitude climate and lower latitudes, with Greenland remaining in a cold stable phase while lower or southern hemisphere latitudes exhibit
significant climatic variations (Barker et al., 2015; Guillevic et al., 2014; Rhodes et al., 2015; Landais et al., 2015). Second, a comparison of several climatic records in Antarctica with the Greenland sequence has highlighted regional differences in the shape of the AIM events and in the phasing between the temperature increase in Antarctica and the abrupt temperature change in Greenland over the long GS's that are associated with a strong discharge in Northern Atlantic, i.e. Heinrich Stadials (Buiron et al., 2012; Landais et al., 2015).
Third, an accurate dating exercise on the west Antarctica WDC ice core over the last 60 ka suggests that the decrease of Antarctic temperature occurs ~200 years after the abrupt temperature increase in Greenland (WAIS Divide Project Members 2015). This has been interpreted as a "northern push for bipolar seesaw" (van Ommen, 2015). Still these studies suffer from the same limitation associated with the ice – air age difference and rely on determination of the past depth of the firn, based on firn densification and/or measurements of $\delta^{15}N$ of $N_2$ in the
trapped air.

Here we use the direct synchronization of NGRIP with EDC, EDML and Vostok based on the $^{10}Be$ records. In Fig. 8 we show the $^{10}Be$-synchronized climate records of the 4 cores, as represented by the δD and $\delta^{18}O$ profiles. Comparing the 3 Antarctic records, one can observe that, while they have the same general features, there are
also significant differences in detail. These are probably due to meteorological effects such as seasonal variation of isotopic ratios, depositional effects, etc. This shows that one must be careful to not over interpret details of individual stable isotope profiles in low accumulation regions of the Antarctic as a climate proxy. This is particularly evident in the Vostok 5G profile, which differs significantly from the previously published profile from the 3G core (Jouzel et al. 1987).

We confirm the Greenland vs. Antarctica relationship observed by Raisbeck et al. (2007) with the increase in $\delta D$ or $\delta^{18}O$ in Antarctica leading the abrupt $\delta^{18}O$ increase in NGRIP by several centuries, which is a classical feature of the bipolar seesaw (Blunier and Brook, 2001), operating here for a small DO (AIM) event. The high resolution of our water isotopes and $^{10}Be$ records also enables us to look at the fine scale Greenland vs. Antarctica temporal relationship around the abrupt warming recorded at NorthGRIP. Although it is impossible to identify by eye any clear inflexion point corresponding to AIM 10 in Vostok, a significant peak of $\delta^{18}O$ is observed at WDC and EDML about 200 years before the abrupt $\delta^{18}O$ increase at NorthGRIP, while the most significant peak in EDC is observed about 150 years after the abrupt Greenland $\delta^{18}O$ increase. The signal is not unambiguous at EDML since two other prominent peaks (but peaking at lower $\delta^{18}O$ values) are also identified in the following centuries. The EDC, WDC and EDML $\delta^{18}O$ profiles also display a $\delta^{18}O$ decrease occurring about 200 years after the NorthGRIP $\delta^{18}O$ increase, as was extensively discussed in WAIS Divide Project Members (2015).

For a proper comparison with the results displayed on WDC in WAIS Divide Project Members (2015), we have used the same statistical approaches, respectively the BREAKFIT software by Mudelsee [2010] and a similar automated routine (referred to as MATLAB routine in Table 2) using a second-order polynomial (rather than a linear for BREAKFIT) fit to the data. The water isotope profiles for EDML and EDC show several wiggles associated with a variability of ~100 − 200 years. Despite this variability, it is still possible to identify a maximum for AIM 10 using the aforementioned statistical tools, keeping in mind limitations linked to the chosen range of detection and short term variability of $\delta^{18}O$ signal. The two statistical tools identify inflexion points on the EDML and EDC $\delta^{18}O$ records. Using both methods, the maximum is reached at EDML, ~210 years before the abrupt Greenland warming, while the maximum detected at Dome C appears to be ~140 years later. To complete this study, using the same statistical tools we have determined the shift between the abrupt Greenland warming and the maximum in the water isotopic record at WDC, and found that the WDC $\delta^{18}O$ maximum is ~142 years earlier. Such analysis leads to the conclusion that all ice cores do not display the same isotopic signal over an AIM, a result in line with previous studies highlighting a strong regional variability at the AIM scale in Antarctica (e.g. Buiron et al., 2012; Landais et al., 2015).

As suggested by a referee (C. Buizert), a stack can be drawn using the 3 $\delta^{18}O$ records from EDML, EDC and WDC (Fig. 8). Because we could not get a $^{10}Be$ synchronization with Match protocol, and unlike EDC and EDML have no independent evidence from volcanic spikes to support our estimated precision, we prefer not to include Vostok in the stack, although tests including it show no significant influence on our conclusions. We emphasize that constructing such a stack is appropriate only if one assumes "a priori" that the "true" climatic part of the $\delta^{18}O$ records for the 3 core locations are synchronous at the studied time scale, which remains to be shown. This stack clearly shows the $\delta^{18}O$ decrease observed 200 years after the abrupt $\delta^{18}O$ increase in Greenland, but also depicts an inflexion point ~200 years (185 - 219 years according to the Matlab routine) before the Greenland abrupt $\delta^{18}O$ increase, so that on the stack, $\delta^{18}O$ displays a ~400 year wide plateau at the time of the abrupt Greenland $\delta^{18}O$ increase. Such a plateau is not unexpected and follows from our observation that in some ice cores (WDC, EDML) the maximum of the AIM was before the abrupt $\delta^{18}O$ increase in Greenland while in another (EDC), the maximum of the AIM occurs after the abrupt Greenland $\delta^{18}O$ increase.

This result seems at first sight to nuance the conclusions presented in WAIS Divide Project Members (2015) using the same approach for the determination of the breakpoint. In both the EDML and WDC ice cores, we find that the maximum in Antarctica for AIM 10 is reached more than 140 years before the Greenland warming. The stack of Antarctic $\delta^{18}O$ records over AIM 10 also suggests that the maximum $\delta^{18}O$ level is reached ~200 years before the Greenland warming. This suggests one must be cautious in drawing a conclusion about a general mechanism for AIM vs. D/O dynamics based on a stack of several cores for a given D/O-AIM event. Similarly, one must be cautious about drawing conclusions based on a stack of different D/O-AIM events in the same core, as proposed in WAIS Divide Project Members (2015). Indeed, the lead / lag of Greenland vs. Antarctica may be different from one event to another. This is evident from Fig. 7 where the maximum of AIM 7, 8 and 10 clearly leads the abrupt warming in Greenland both at WDC and EDML. Both stacking several AIM together on one core or stacking one AIM using several cores can be misleading when discussing short term variability that may be different from one core to another, or one event to another.

Finally, even if the location of the maximum of the AIM, hence the statement of "northern push for bipolar seesaw", can be disputed, we confirm that the main $\delta^{18}O$ decrease in Antarctica occurs ~200 years after the Greenland temperature increase. This particular result is robust in different Antarctic sectors and for different DO/AIM events and should be retained to refine our understanding of bipolar seesaw.

**7 Conclusion**

We have shown here that in periods where there are significant production variations of $^{10}Be$, it is possible to synchronize Greenland and Antarctic ice cores with decadal precision. This in turn means that the climate and other environmental parameters registered in these ice cores can be synchronized at this same precision, thus allowing different models and mechanisms to be more finely tested.

**Acknowledgments**

This project was initiated during discussions between the first author and Sigfus Johnsen at an EPICA meeting at San Feliu, Spain, in 2003. Sigfus was always a strong supporter of 10Be measurements in ice cores, and enthusiastically responded positively to the idea of using high resolution NGRIP samples to overcome limitations encountered with GRIP samples. We therefore dedicate this paper to Sigfus' memory. We thank C. Buizert and an anonymous reviewer for helpful comments. This work could not have been achieved without the many persons involved in logistics, drill development and implementation, ice core processing and analysis, that we would like to acknowledge. This includes North GRIP directed and organized by the Department of Geophysics at the Niels Bohr Institute for Astronomy, Physics and Geophysics, University of Copenhagen and supported by funding agencies and institutions from participating countries, and EPICA, the European Project for Ice Coring in Antarctica, a joint European Science Foundation/European Commission (EU) scientific program, funded by the EU and by national contributions from Belgium, Denmark, France, Germany, Italy, The Netherlands, Norway, Sweden, Switzerland, and the U.K. We thank all Russian and French participants of the Vostok drilling, field work and ice sampling. The research leading to these results has received funding from the European Research Council under the European Union's Seventh Framework Programme (FP7/20072013)/ERC grant agreement no. 30604

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

| Event | Depth NGRIP (m) | GICC05 age (yrs b2k) | depth EDML observed (m) | Depth EDML Predicted[3] (m) | depth EDC observed[1] (m) | Depth EDC predicted[3] (m) | depth Vostok 5G (m) |
|---|---|---|---|---|---|---|---|
| BeA | 2106.01 | 40563 | 1362.28 | | 731.65 | | 598.16 |
| BeB | 2109.62 | 40794 | 1366.56 | | 734.55 | | 600.75 |
| L1 | 2111.58 | 40912 | 1369.54 | 1368.88 | | 735.94 | |
| BeC | 2113.22 | 41002 | 1370.57 | | 736.97 | | 603.99 |
| L2 | 2115.41 | 41109 | 1372.73 | 1372.22 | 738.19[2] | 737.92 | |
| L3 | 2118.62 | 41249 | 1375.15 | 1374.71 | 739.80[2] | 739.44 | |
| BeD | 2129.82 | 41858 | 1386.61 | | 746.68 | | 614.86 |
| BeE | 2132.64 | 42067 | 1390.49 | | 749.17 | | 617.40 |

[1]EDC 96 depth

[2]from Svensson et al. (2013) using EDML-EDC correlation of Severi et al. (2007)

[3]from $^{10}$Be synchronization (Fig 3b,c)

**Table 1: In the first two columns, are reported the depth and GICC05 ages (Svensson et al. 2008) of our $^{10}$Be tie points (BeA–BeE) and the volcanic markers (L1-L3) of Svensson et al. (2013) at the NGRIP site (see text). The following columns provide the depths of corresponding depths for EDML, EDC and Vostok ice cores either observed or predicted.**

| Age difference of the inflexion point with respect to the mid-slope of abrupt warming in NGRIP | MATLAB algorithm (WAIS-Divide Project Members, 2015) | BREAKFIT software (Mudelsee, 2010) |
|---|---|---|
| EDML (this study) | -213 ± 40 years | -207 ± 50 years |
| EDC (this study) | +130 ± 30 years | +150 ± 50 years |
| Vostok (this study) | Non significant | Non significant |
| WDC, WAIS-Divide Project Members (2015) | -156 ± 50 years | -129 ± 50 years |

**Table 2: Phasing between NGRIP GS – GIS transition and AIM for different Antarctic ice cores over DO-AIM 10.**

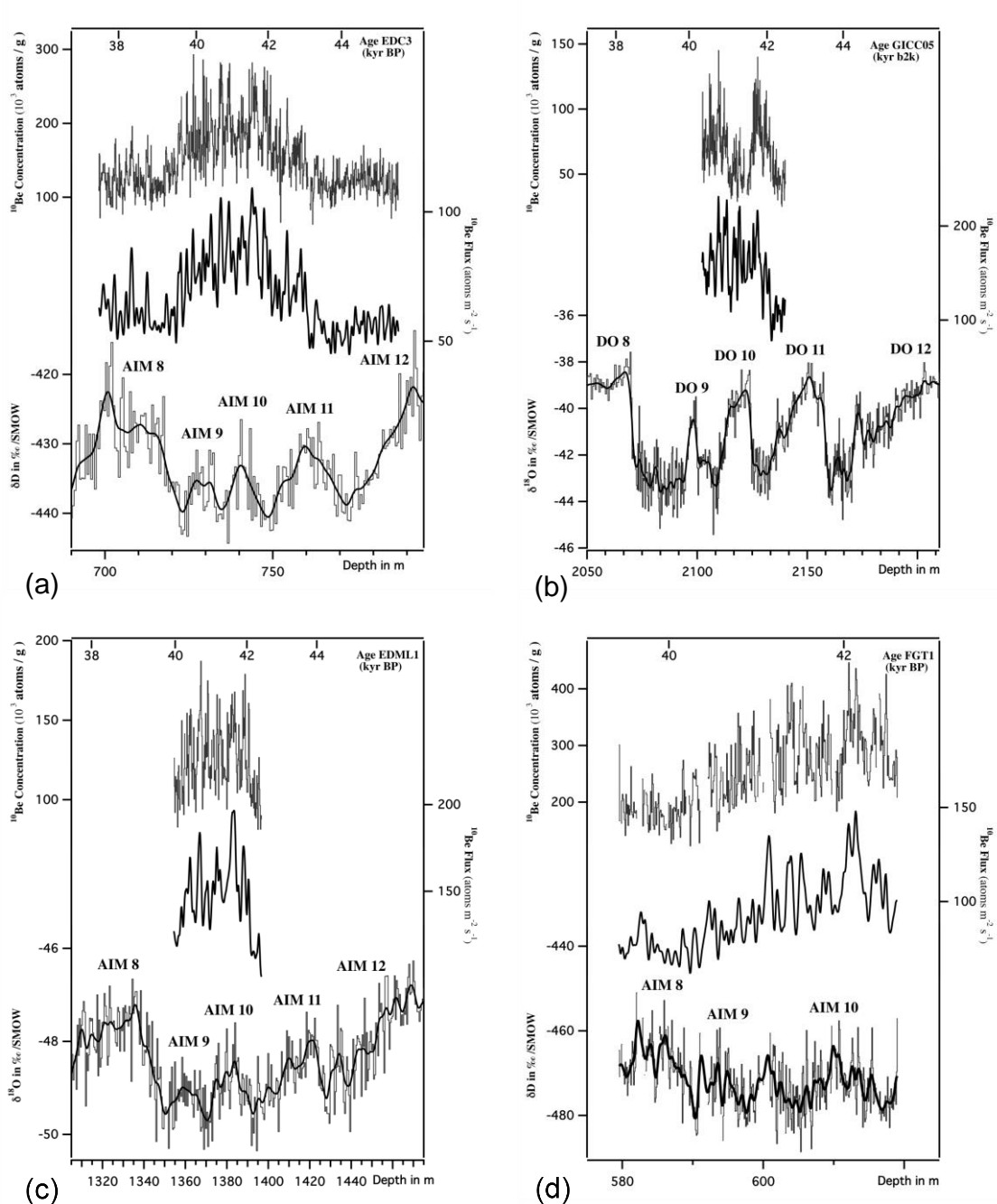

**Figure 1:** $^{10}$Be concentrations, $^{10}$Be flux and stable isotope profiles ($\delta$D or $\delta^{18}$O) and their running averages as a function of depth and age in 4 ice cores a) EPICA Dome C; EDC3 timescale (Parrenin et al., 2007), b) NGRIP; GICC05 timescale (Andersen et al., 2006), c) EDML; EDML1 timescale (Ruth et al., 2007), and d) Vostok: FGT1 timescale (Parrenin et al., 2004). The $^{10}$Be flux has been calculated using accumulation rates used in the above timescales and smoothed with a 50 year binomial filter. AIM stands for Antarctic Isotope Maximum.

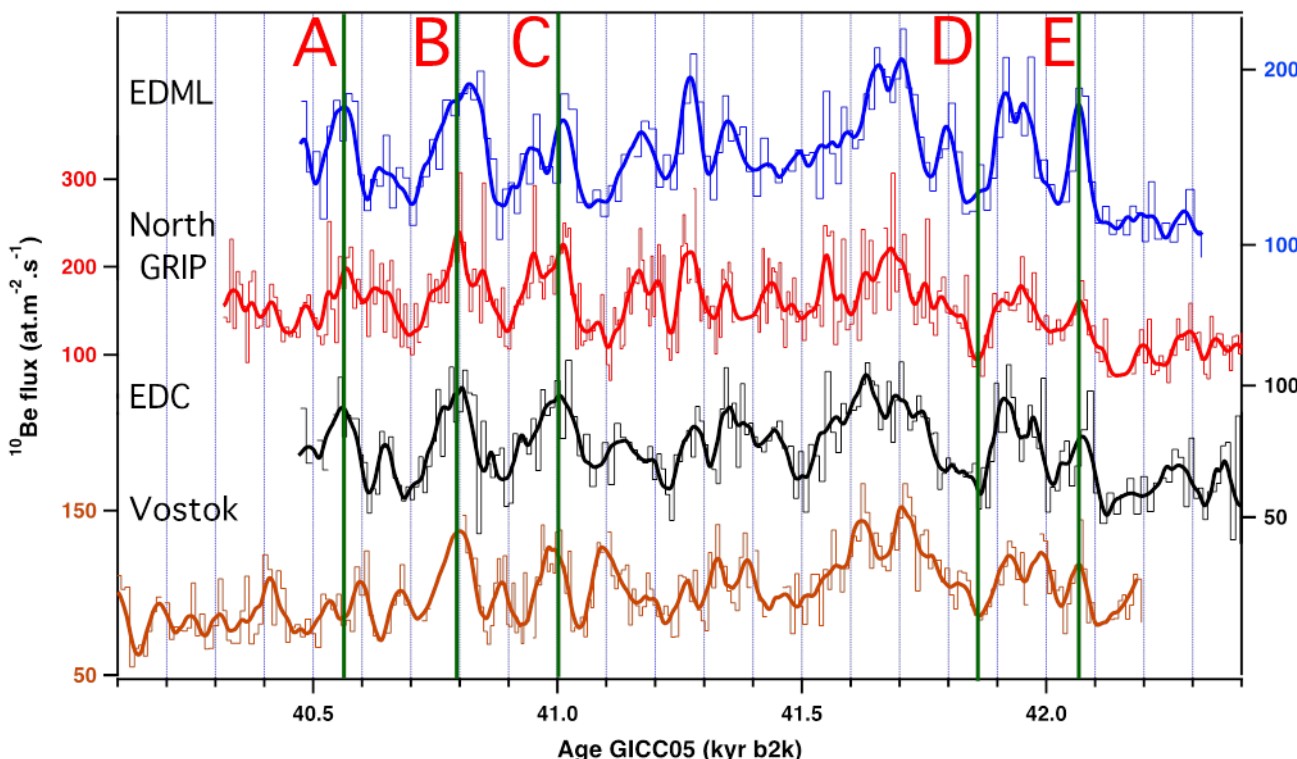

**Figure 2: Beryllium 10 fluxes after synchronisation to NGRIP (GICC05) using Match protocol (Lisiecki and Lisiecki, 2002) for EDC and EDML, or using 5 tie-points (A-E) and Analyseries (Paillard et al., 1996) for Vostok. For each site the detailed fluxes (histograms calculated using 11 cm accumulation rates for NGRIP, 55 cm for EDC, 50 cm for EDML and 10 cm for Vostok) and results smoothed with a 50 year binomial filter (curve) are shown.**

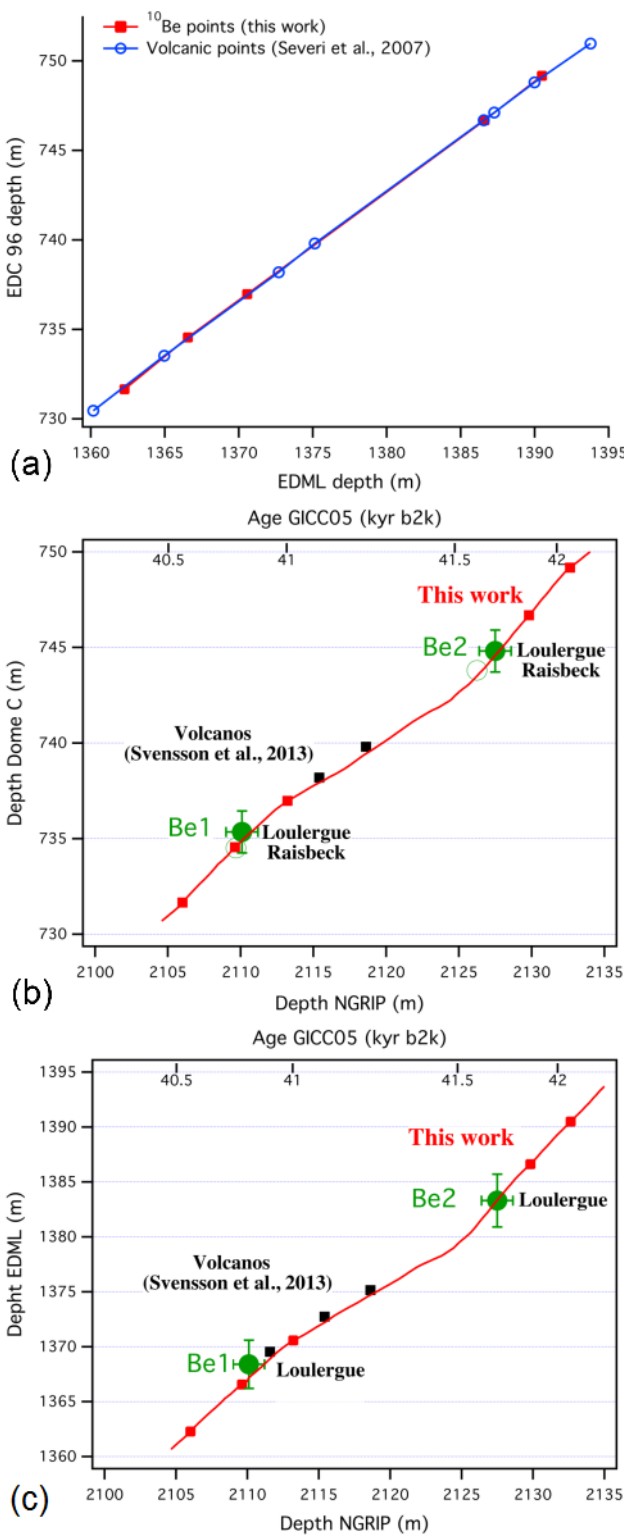

**Figure 3: Comparison of the different tie points between EDC and EDML (a) NGRIP and Dome C (b) and NGRIP and EDML (c) as derived from beryllium-10 from this work (red squares and line), from Loulergue et al. (2007) (green full circles) from Raisbeck et al. (2007) (green empty circles) with in addition from EDML/EDC volcanic tie points (open blue circles from Severi et al., 2007) and proposed bipolar volcanic points (full black squares) (Svensson et al., 2013). We have indicated the uncertainty for the tie points used in Loulergue et al. (2007). For the new ¹⁰Be and volcanic tie points, the uncertainties are smaller than their symbols.**

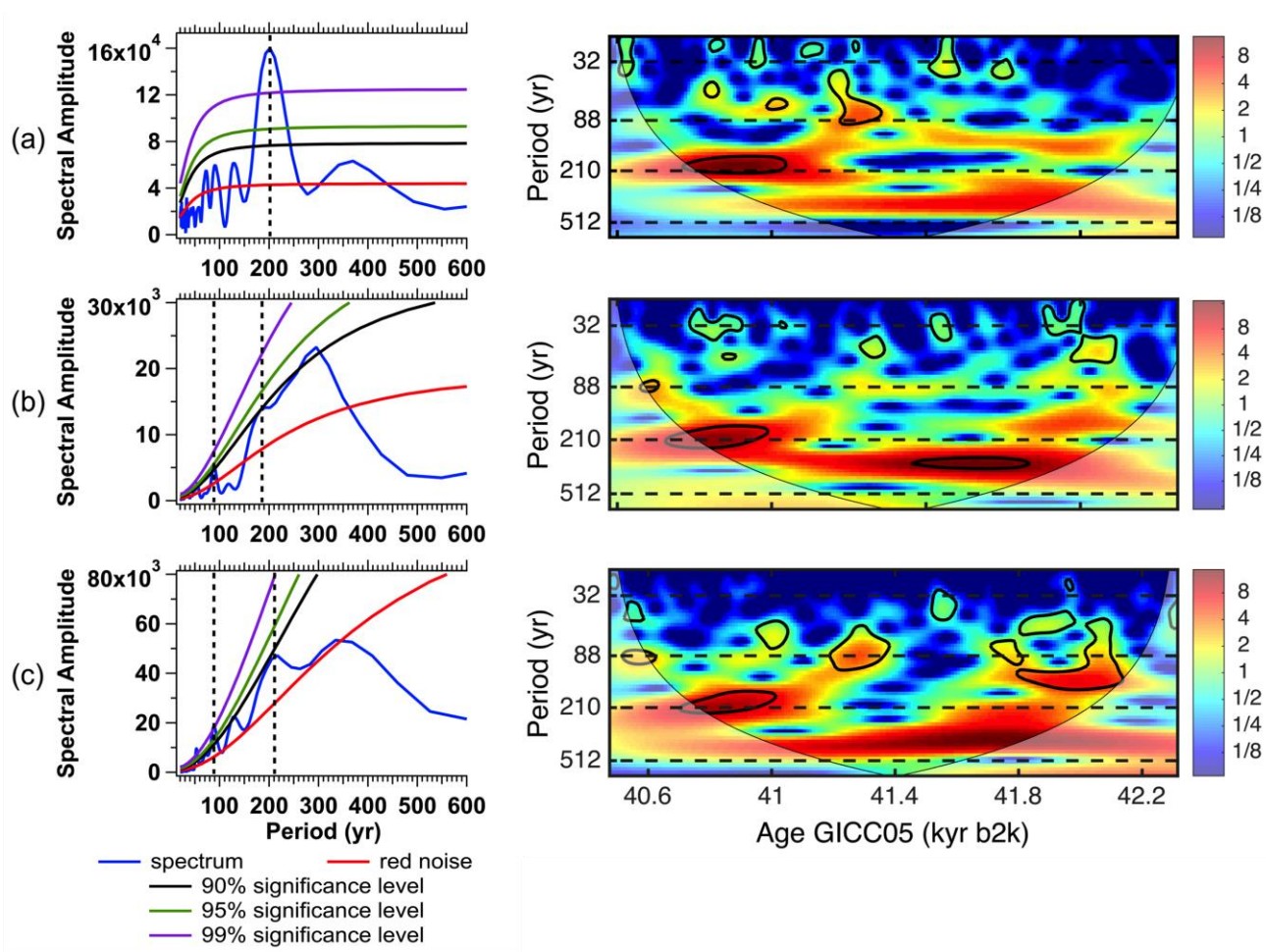

**Figure 4: Fourier (left) and wavelet (right) analyses of our [10]Be flux records at (a) NGRIP, (b) EDC and (c) EDML from 40.480 to 42.320 kyr b2k on the GICC05 age scale. The solid black lines on the wavelet panels indicate the regions which are significant at 95% level.ion**

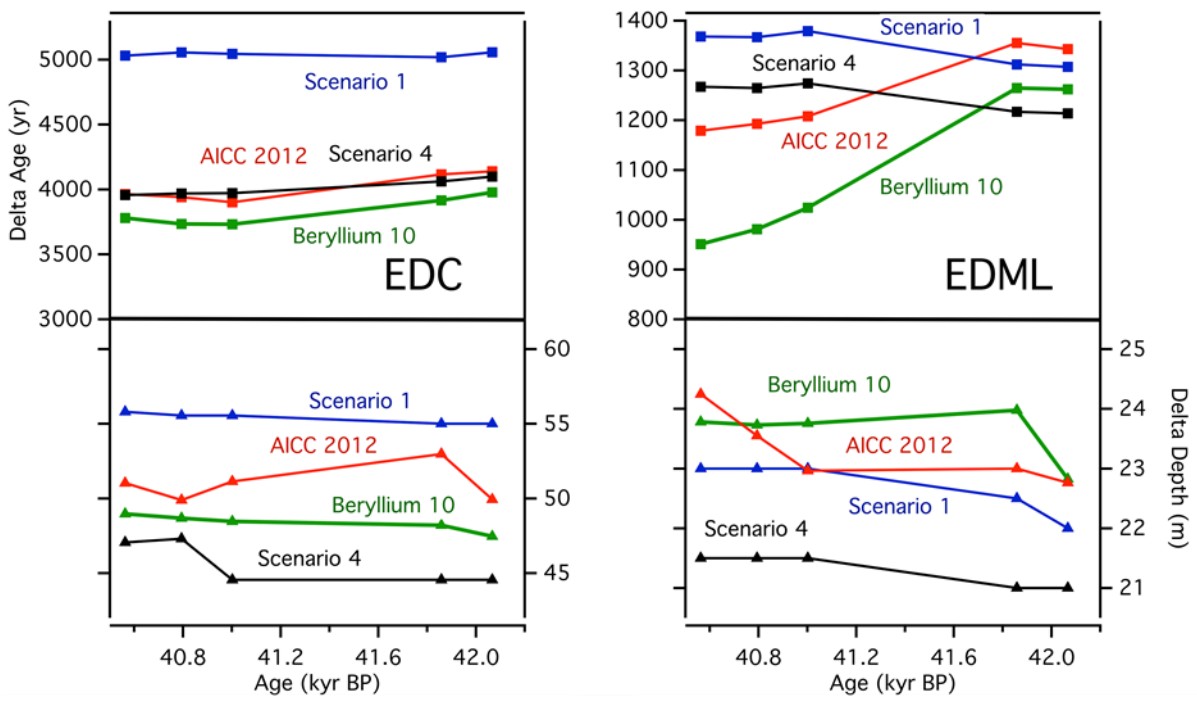

**Figure 5: Δage and Δdepth derived for the 5 beryllium-10 tie points derived in this work compared with their values as used in scenario 1 and 4 (Loulergue et al., 2007) and for AICC2012 (Bazin et al., 2013).**

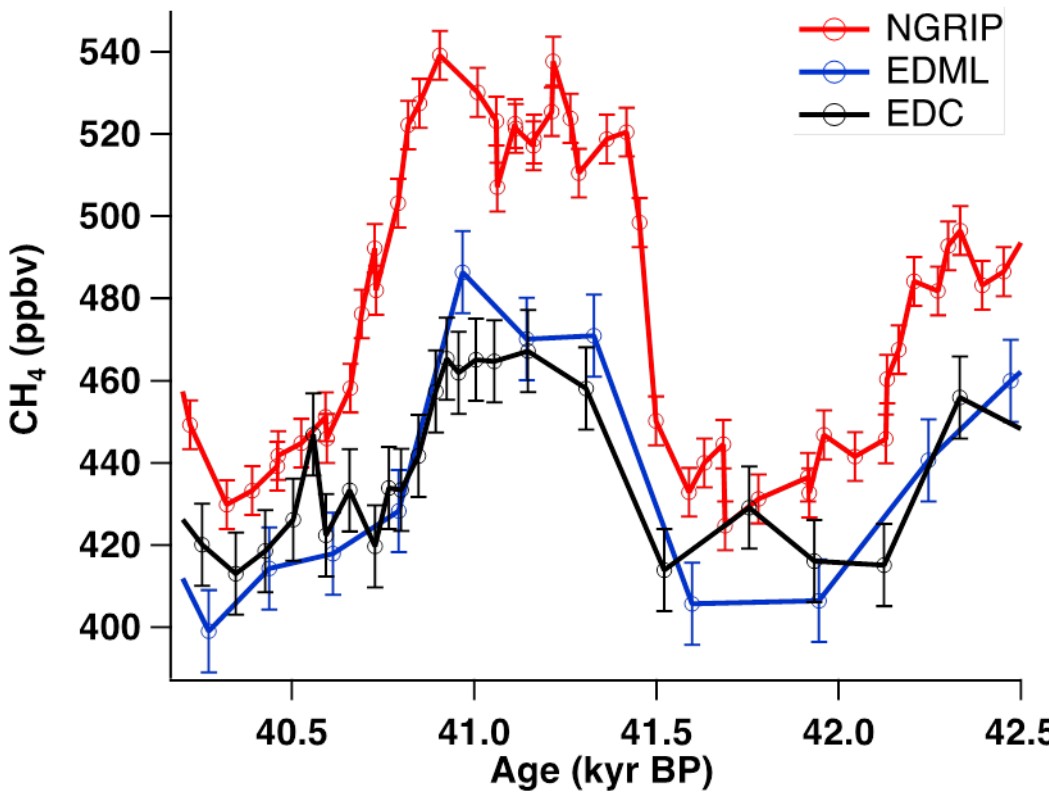

**Figure 6: Synchronization of the methane profile from NGRIP with EDC and EDML profiles. The methane data are from Baumgartner et al. (2014) for NGRIP, Loulergue et al. (2007) for EDC and Schilt et al. (2010) for EDML.**

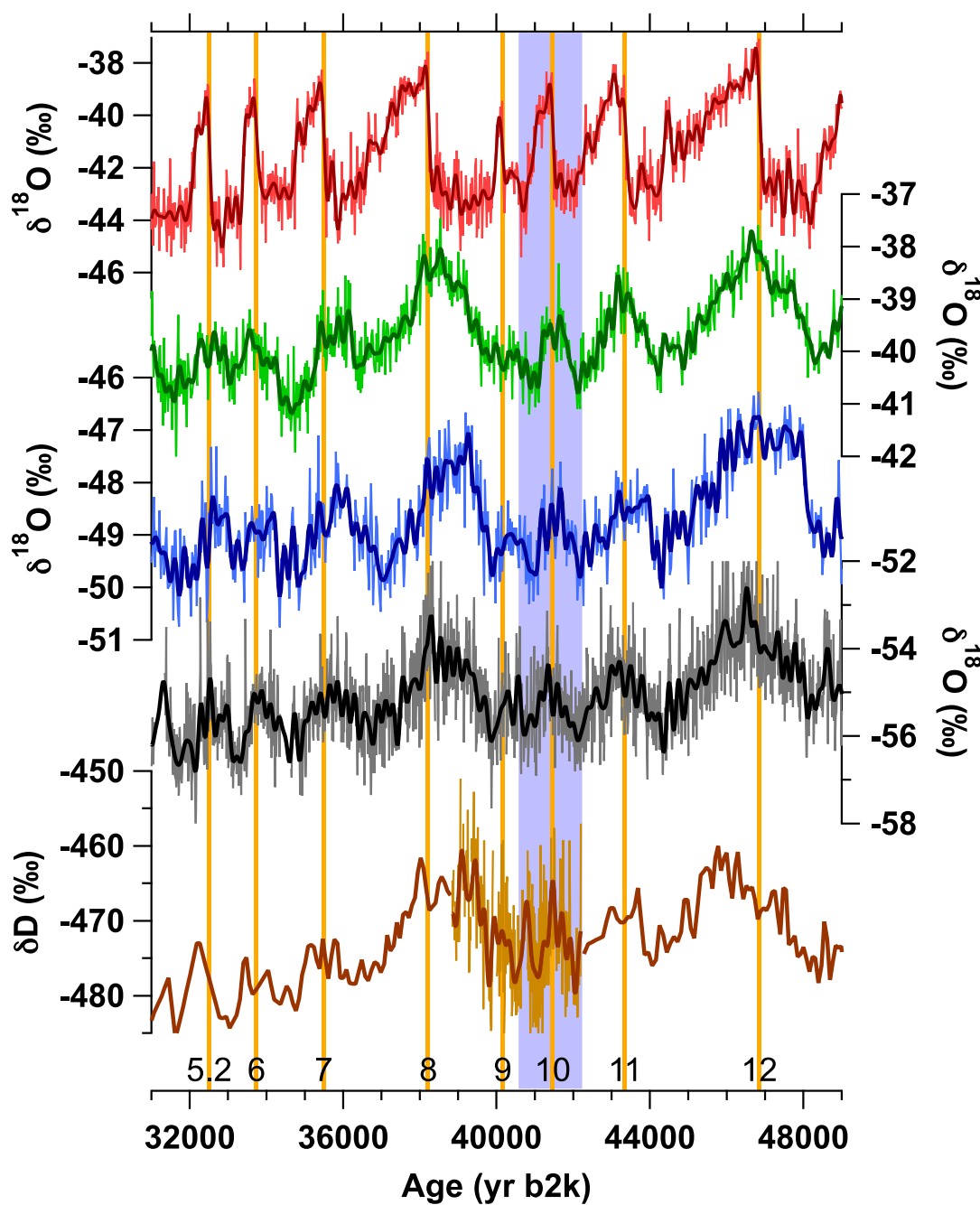

**Figure 7: Water isotopic records for NGRIP and 4 Antarctic ice cores over DO events 5 to 12 on the GICC05 timescale with the transition between GS and GIS indicated by the vertical orange lines. The synchronization between NGRIP (red), EDML (blue), EDC (black) and Vostok (brown) is based on the AICC2012 timescale while the WDC (green) record has been transferred on the GICC05 timescale as WD2014 in Buizert et al. (2015).**

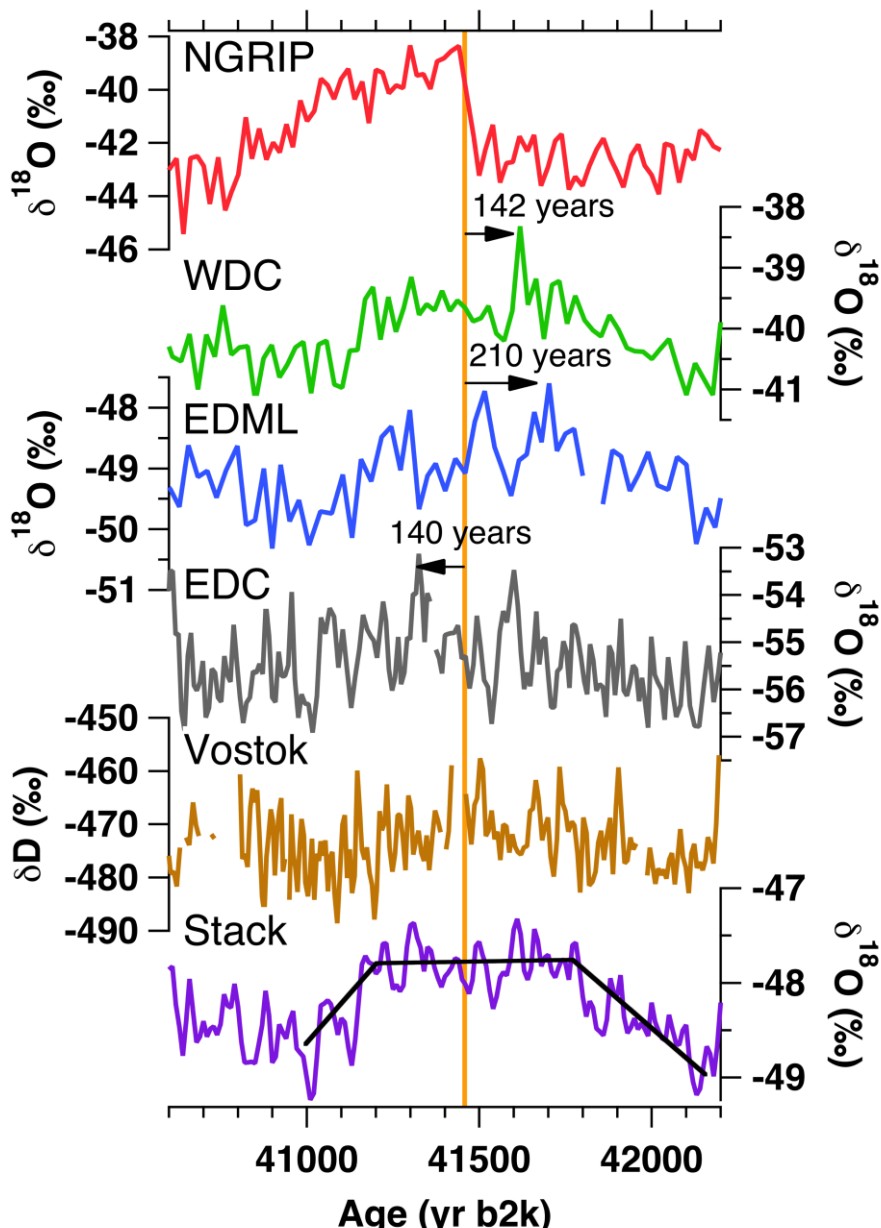

**Figure 8: Phasing between the GS-GIS transition in Greenland (vertical orange line) and the maximum of the AIM 10 in 4 different Antarctic ice cores on the GICC05 age scale. The synchronicity between Vostok, EDC, EDML and NGRIP is based on the $^{10}$Be records while the WDC isotopic record has been drawn on the GICC05 timescale using the correspondence between the WSD timescale and GICC05 given in Buizert et al. (2015). A stack (purple curve) is drawn using the 3 $\delta^{18}$O records from EDML, EDC and WDC, and the black lines correspond to its inflexions identified by eye.**