# Peer review of "An improved North-South synchronization of ice core records around the 41 kyr beryllium-10 peak"

_Climate of the Past, 2016_

## Referee Comment (RC1) · C. Buizert (Referee) · 3 Aug 2016

Review of Raisbeck et al. by Christo Buizert

Raisbeck et al. present new high-resolution [10]Be records for several ice cores, which allows for the most precise interpolar ice core synchronization to date during MIS 3. The authors use this synchronization to (1) estimate the spectral properties of [10]Be variations, (2) investigate the $\Delta$age/$\Delta$depth evolution in Antarctic ice cores, and (3) investigate the phasing of the bipolar seesaw. All three problems are very relevant to the ice core and paleoclimate communities, making this paper a valuable contribution. Overall the paper is clear and well written.

Even though I disagree with the authors' interpretation of the bipolar seesaw phasing (see below), I fully recommend this work for publication in Climate of the Past. Disagreement on the interpretation of data is a normal and healthy part of science, and it does not detract from the main contribution of this work (which is the high-precision [10]Be synchronization).

Comments:

(1) My main concern is the author's interpretation of the bipolar seesaw phasing (section 6). In my experience, it is not meaningful to investigate the phasing of a single AIM event in a single core, because the climatic seesaw signal is overwhelmed by high-frequency $\delta^{18}O$ variability due to weather, deposition and other local events. The authors clearly demonstrate this point for AIM 10. However, once the signal from several AIM events is averaged, the shared climatic seesaw signal is clearly revealed. This averaging strategy was used in WAIS Divide project members (2015) – I will refer that paper as WDPM15.

To demonstrate this principle, I averaged the [10]Be-synchronized AIM 10 event in the WD, EDML and EDC $\delta^{18}O$ time series (see figure R1 below). I synched the EDML record myself using Table 1, and Grant Raisbeck kindly provided the synched high-res EDC record. I did not have access to the VK record. The resulting average is plotted at the bottom of the figure (orange), on top of the WAIS Divide AIM3-18 stack from WDPM15 (purple). It is clear that the multi-core AIM 10 average agrees well with the WDPM15 stack, and shows a clear cooling trend some ~200 years after the abrupt DO 10 warming event. The AIM 10 stack is more noisy than the WAIS stack simply because it averages over fewer events.

Landais et al. (2015) and other papers have shown clearly that AIM events are expressed differently at various sites, and I do not dispute that. However, it is also clear that whenever several AIM events are averaged to improve the signal-to-noise ratio, the ~200 year time scale shows up (it is also visible in many individual AIM events). This timescale must tell us something about the climatic coupling between the hemispheres. The peak $\delta^{18}O$ value for a given AIM event at an individual core site can of course be different from this 200 year lag, due to local high-frequency weather and deposition effects. Interpreting only the position of the $\delta^{18}O$ -maximum is therefore too simplistic, in my view. At EDML the AIM events seem to have a flat top, as opposed to the more triangular shape at e.g. Byrd, WAIS, EDC and TALDICE. However, adding EDML to the multi-core averaging does not seem to alter the fact that there is substantial (Antarctic-wide?!) cooling 200 years after abrupt NH warming (my Fig. R1 below).

I request that in a revised MS the authors include a multi-core AIM 10 average (i.e. average of WDC, EDC, EDML and VK) in Fig. 7, and discuss some of the points I made above. I do not think our interpretations are mutually exclusive. On average, there is substantial Antarctic cooling 200 years after NH warming (my preferred interpretation), yet the $\delta^{18}O$ isotopic maximum during a single AIM event in a single core can differ substantially from this 200 year delay due to local effects (the authors' interpretation). I trust that the authors are willing to present both interpretations side by side. I think this will make the bipolar seesaw phasing less confusing to readers (who may be familiar with WDPM15), and make the paper overall more robust.

(2) Section 2 describes the new NGRIP, EDML and VK records in detail, but not EDC. Please add a few lines describing the EDC $^{10}$Be record also.

(3) In section 3 the authors test the accuracy of their synchronization. The $^{10}$Be also links the EDML and EDC cores, which can be directly compared to the volcanic synchronization of Severi et al 2007. This would provide a true test of the uncertainty in the $^{10}$Be synchronization, given that volcanic matching is the gold standard of synchronization. I tried to do this (see Fig. R2), and found a small, but constant offset between the synchronizations which is ~ 70 cm on EDC / ~110 cm on EDML. (I took the Be ties from Table 1, and the volcanic links (on EDC99) from the AICC2012 documentation).

Do you have any idea where this offset could come from? I would urge the authors to double check for trivial mistakes such as converting bag numbers to depths, or similar. Or is this the offset between the EDC96 and EDC99 cores? I could not find any information on which EDC core was used. In either case, the direct comparison to Severi et al. (2007) provides a great opportunity to test the precision of the $^{10}$Be ties. It may be worth including this comparison as a third panel to Fig. 3.

(4) In section 5 I am a confused by the different trends in the $\Delta$age and $\Delta$depth. In my mind the two are exchangeable, as you can calculate one from the other using the ice chronology. For example, how is it possible that the EDC $\Delta$age for AICC2012 and Scen4 (red and black) are identical, while their $\Delta$depth is so different? Doesn't that imply that these two chronologies have completely different annual layer thickness (while both are synched via $^{10}$Be)?

The authors could provide a few more details on how the $\Delta$age and $\Delta$depth are constructed, which may help in understanding what's going on. For example, which chronology is used for the 2 Loulergue scenarios? I would assume the authors ran the densification models using the AICC2012 T, Acc and chronology for consistency?

(5) In section 6 (P7 L25-27) the authors use the BREAKFIT routine and a MATLAB routine to estimate the breakpoint in the data. I don't think these routines are particularly fit for the problem at hand, given that the time series are short and very noisy. The data range must be picked to isolate AIM 10, and then the routines require the user to specify a range where they believe the breakpoint is located. Because it is short and noisy, these subjective choices seem to matter a lot for AIM 10. For example, I tried the fitting routine for just AIM 10 at WAIS (where I did this before), and got a timing of -10 or +205 years depending on whether I used linear or 2$^{nd}$ order fitting. I don't mean to suggest that the authors applied

the code incorrectly, I simply want to highlight that for this particular problem the outcome is very sensitive to the subjective choices of the operator. The authors may have had the same experience.

For longer time series with less noise the routines perform well, and become independent of the subjective choices of the user.

The isotopic maxima that the authors identify in Fig. 7 can also be picked out by eye, so I suggest the authors just remove the fitting routines from the paper (my preference) or provide more details on how the fitting routines were applied (data range, etc) and how the uncertainties were estimated.

Minor/language:

Throughout the text: I would suggest replacing "delta age" and "delta depth" with $\Delta$age and $\Delta$depth (i.e. using Greek Delta symbol) to confirm with common usage in the ice core literature

Throughout: WAIS Divide is spelled without a hyphen between "WAIS" and "Divide"

Title: "41 k" should probably be changed to "41 kyr/ka". Also, please include a hyphen in "beryllium-10".

P1 L9: are these 2sigma uncertainty values?

P1 L 20: Remove "our". The author lists of Raisbeck et al. 2007 and Raisbeck et al. 2016 are not identical.

P1 L 21: ... estimates of the DEPTH difference between ...

P1 L25: In a previous study, Raisbeck et al. (2007) have ....  (same reason as above)

Section 3: Please specify confidence intervals for the uncertainty estimates. Are these 2 sigma?

Section 3: Maybe note that the precision on the WAIS Divide $CH_4$ interpolar synchronization at 40ka is estimated to be +/- 73 years (2 sigma uncertainty in $\Delta$age; see Buizert et al. 2015 Fig. 3e), and therefore the new $^{10}$Be synchro is more precise.

P5 L12-14: Do you mean meteorology interferes with the actual atmospheric $^{10}$Be production rate, or do you mean it leads to differences in deposition, transport and dilution? Because of the annual layer count, Greenland Acc can be reconstructed much more accurately that Antarctic Acc – could this be one of the reasons the 200yr peak is better resolved at NGRIP?

P7 L21-22: I think it would be good to cite e.g. Blunier and Brook 2001 here, who were among the first to describe the asynchronous coupling clearly.

Figs 7 and 8, caption: please specify how the WD2014 chronology was transferred to GICC05. I assume you divided by 1.0063 and then added 50 years to get from BP1950 to B2k?

[Figure]

**Figure R1**

[Figure]

---

## Referee Comment (RC2) · Anonymous Referee #2 · 23 Aug 2016

Raisbeck et al. present new high-resolution 10Be data (NGRIP, EDML & Vostok ice cores) that they use to synchronise ice core records from Greenland and Antarctica over the period of the Laschamp geomagnetic field minimum. They discuss the precision of this synchronisation and the phasing of water-isotope variations around D/O 10. In addition, they investigate cyclicities in the 10Be records and make an assessment of the ice age - gas age difference around this period.

I think this is a very interesting paper that shows new important 10Be records. I certainly recommend the publication of the manuscript in Climate of the Past after the authors addressed several important remarks that I have listed below. I also think that the publication of the paper should be linked to the requirement of publishing the 10Be

data as a supplementary table so that the analyses can be repeated with different methods.

Comments: Regarding the flux calculation there are three important comments that need to be addressed: 1: On Page 3, line 10 the authors say that the NGRIP accumulation rate is based on the ss09sea time scale. However, figure caption 1 says that the accumulation rate is based on the respective time scales (GICC05 for NGRIP). So there is a contradiction that needs to be sorted out. I think GICC05 should be used as time scale and as the basis for the accumulation rate. 2: I do not understand that the flux calculations are based on a smoothed version of the accumulation rates (see page 3, line 8). This can introduce jumps in the 10Be flux at transitions (e.g. 10Be concentrations react immediately while a smoothed accumulation record follows slowly leading to artificial jumps in the fluxes). The flux calculation needs to be done on the raw data. Smoothing can then be done afterwards on the flux record. 3: It is not a priori clear that the 10Be flux indeed reflects the 10Be production rate only. It could have imprints of climate change (e.g. during D/O transitions). Therefore, the authors should discuss if the peaks they used for the synchronisation are robust, i.e. independent of flux calculation (also visible in the 10Be concentrations).

I do not want to question the synchronisation results. However, they seem to critically depend on the choice of the fix points. As visible in figure 2 there might be alternative choices for fix points due to the cyclic behaviour of the 10Be data. I would like that the authors discuss in more detail how they have chosen the fix points, how robust these choices are and if it could be possible that other choices could be done. If there are alternative choices it could influence the subsequent discussions. As an example, I would like that the authors to discuss the 3 youngest peaks in the Vostok data. The youngest peak is clearly not there. Could it be possible to shift the whole record 200 years younger which might lead to a better agreement of the peaks in the younger part of the record. This could be discussed in connection with the MATCH routine i.e. what are the reasons to exclude such scenarios. The authors mention the missing data in

the Vostok record. However, for the periods of disagreement missing data does not seem to be the major problem.

I think the errors for the synchronisation are not well defined. If I understand correctly they are based on the agreement of the 2 synchronisation methods. However, outliers can shift peaks and this could lead to biases in all synchronisation methods i.e. leading to a false sense of confirmation. I recommend that the author treat synchronisation errors with caution. An indication for systematic problems might be the systematic offset of 10Be-synchronised time scales and the identification of common volcanic-induced spikes.

It is never discussed how the synchronisation errors should be understood. Does an error of <20 year correspond to a 1 sigma error or a 2 sigma error?

I think the spectral analysis is interesting but it shows that the results of Fourier analysis (especially significance levels) depend very sensitively on noise estimates (it is interesting that the noise levels are very different in the different records leading to the different results on the significance of cycles). The wavelet analysis and the agreement of the different records indicate that there are common cycles irrespective of the significance analyses (otherwise the synchronisation would not work. . .).

I think the authors should not attribute possible uncertainties to meteoric influences on 10Be records from Antarctica only (page 5 line 11), or do the authors want to imply that the Greenland data does not contain any meteorological influences?

I have question regarding the approach for the methane synchronisation and delta age calculation. The authors synchronise the whole section (figure 5) but not single peaks. However, looking at methane from NGRIP one gets the impression that the data in the youngest period is placed rather too young. Can this explain the delta age difference between the 10Be and other methods shown in figure 6.

I appreciate figure 7 since I think it shows the data in a honest way. It also shows that,

in my opinion, the results from the applied statistical tools needs to be treated with caution. For example, EDC shows a similar early isotope spike as WDC and therefore it appears to me that the lead/lag discussion is very much influenced by the noise in the data and the applied statistical tools. These uncertainties could be emphasised even more.

Regarding figure 8 and the discussion on longer time scales. I understand that this discussion fits into the discussion following figure 7. However, to me it feels like a step backwards. The authors have this great 10Be data and the synchronisation and then they go back to the results from an older synchronisation. Maybe figure 8 would fit better as figure 1 in connection to the introduction of the general topic.

It is not 100% clear to me on what the errors in table 2 are based on. Uncertainties in the time scale synchronisation and/or in the method to find the transition points?

Details:

I think the introduction is quite short but OK. I was also wondering why the authors list the results in the introduction.

Page 2 line 19: Did the Vostok samples really weigh >500g? It seems like quite a lot even considering that the measurements were done a while ago.

page 7 line 12: I would say "10Be-synchronised climate records"

In general, I think the paper is very well written!

———————————————————

---

## Author Comment (AC1) · 26 Oct 2016

Review of Raisbeck et al. by Christo Buizert

Raisbeck et al. present new high –resolution 10Be records for several ice cores, which allows for the most precise interpolar ice core synchronization to date during MIS 3. The authors use this synchronization to (1) estimate the spectral properties of 10Be variations, (2) investigate the Δage/Δdepth evolution in Antarctic ice cores, and (3)investigate the phasing of the bipolar seesaw. All three problems are very relevant to the ice core and paleoclimate communities, making this paper a valuable contribution. Overall the paper is clear and well written.

Even though I disagree with the authors' interpretation of the bipolar seesaw phasing (see below), I fully recommend this work for publication in Climate of the Past. Disagreement on the interpretation of data is a normal and healthy part of science, and it does not detract from the main contribution of this work (which is the high-precision 10Be synchronization).

**We thank Christo Buizert for his comments, particularly the relevance of the paper with regard to phasing of the climate signal. Our replies are given below.**

Comments:

(1) My main concern is the author's interpretation of the bipolar seesaw phasing (section 6). In my experience, it is not meaningful to investigate the phasing of a single AIM event in a single core, because the climatic seesaw signal is overwhelmed by high-frequency δ18O variability due to weather, deposition and other local events. The authors clearly demonstrate this point for AIM 10. However, once the signal from several AIM events is averaged, the shared climatic seesaw signal is clearly revealed. This averaging strategy was used in WAIS Divide project members (2015) – I will refer that paper as WDPM15. To demonstrate this principle, I averaged the 10Be-synchronized AIM 10 event in the WD, EDML and EDC δ18O time series (see figure R1 below). I synched the EDML record myself using Table 1, and Grant Raisbeck kindly provided the synched high-res EDC record. I did not have access to the VK record. The resulting average is plotted at the bottom of the figure (orange), on top of the WAIS Divide AIM3-18 stack from WDPM15 (purple). It is clear that the multi-core AIM 10 average agrees well with the WDPM15 stack, and shows a clear cooling trend some ~200 years after the abrupt DO 10 warming event. The AIM 10 stack is more noisy than the WAIS stack simply because it averages over fewer events.

Landais et al. (2015) and other papers have shown clearly that AIM events are expressed differently at various sites, and I do not dispute that. However, it is also clear that whenever several AIM events are averaged to improve the signal-to-noise ratio, the ~200 year time scale shows up (it is also visible in many individual AIM events). This timescale must tell us something about the climatic coupling between the hemispheres. The peak δ18O value for a given AIM event at an individual core site can of course be different from this 200 year lag, due to local high-frequency weather and deposition effects. Interpreting only the position of the δ18O -maximum is therefore too simplistic, in my view.

At EDML the AIM events seem to have a flat top, as opposed to the more triangular shape at e.g. Byrd, WAIS, EDC and TALDICE. However, adding EDML to the multi-core averaging does not seem to alter the fact that there is substantial (Antarctic-wide?!) cooling 200 years after abrupt NH warming (my Fig. R1 below). I request that in a revised MS the authors include a multi-core AIM 10 average (i.e. average of WDC, EDC, EDML and VK) in Fig. 7, and discuss some of the points I made above. I do not think our interpretations are mutually exclusive. On average, there is substantial Antarctic cooling 200 years after NH warming (my preferred interpretation), yet the δ18O isotopic maximum during a single AIM event in a single core can differ substantially from this 200 year delay due to local effects (the authors' interpretation). I trust that the authors are willing to present both interpretations side by side.

I think this will make the bipolar seesaw phasing less confusing to readers (who may be familiar with WDPM15), and make the paper overall more robust.

**It should indeed be noted that we were not disputing in this manuscript the fact that the main cooling in Antarctica followed by almost 200 years the abrupt warming in Greenland. What we want to highlight is the fact that the warming trend in several Antarctic records (EDML and WAIS) is interrupted before the abrupt Greenland temperature increase and thus that there is not one single inflexion point in the Antarctic temperature water isotopic records during the AIM.**
**It thus seems that we agree on the main point which is the cooling. However, we believe that the regional variability during the warming phase should also be highlighted because it also has important implications for the mechanisms and questions the affirmation of "Northern push for bipolar seesaw". We also feel that before making a stack, we should document the variability from one site to another and from one AIM to another. This is especially important when discussing small phase lags of about 200 years.**
**In the revised version, we will present the stack for the isotope records of the ice cores WDC, EDML and EDC over AIM 10 as suggested, while noting that this is only appropriate if one assumes " a priori" that climate changes simultaneously over all Antarctica. The Vostok isotopic record has not been included in the stack because we could not get a $^{10}$Be synchronization with Match protocol, and unlike EDC and EDML, we do not have the independent evidence from volcanic spikes to support our estimated precision.**
**Two important slope changes can be identified on the stack (either by eye or using one of the routines described in the main text): the first one occurs before the rapid warming in Greenland and the second one occurs after the rapid warming in Greenland. The second breakpoint is indeed coherent with the main cooling in Antarctica occurring 200 years after the rapid warming in Greenland.**
**The revised version will thus describe the regional variability from one core to the other and include a discussion of the stacked record and the two slope changes.**

(2) Section 2 describes the new NGRIP, EDML and VK records in detail, but not EDC. Please add a few lines describing the EDC 10Be record also.

**This was because the EDC record was described in detail in the Raisbeck et al. (2007) paper. We will add a sentence to that effect in the revised version.**

(3) In section 3 the authors test the accuracy of their synchronization. The 10Be also links the EDML and EDC cores, which can be directly compared to the volcanic synchronization of Severi et al 2007. This would provide a true test of the uncertainty in the 10Be synchronization, given that volcanic matching is the gold standard of synchronization. I tried to do this (see Fig. R2), and found a small, but constant offset between the synchronizations which is ~ 70 cm on EDC / ~110 cm on EDML. (I took the Be ties from Table 1, and the volcanic links (on EDC99) from the AICC 2012 documentation).
Do you have any idea where this offset could come from? I would urge the authors to double check for trivial mistakes such as converting bag numbers to depths, or similar. Or is this the offset between the EDC96 and EDC99 cores? I could not find any information on which EDC core was used. In either case, the direct comparison to Severi et al. (2007) provides a great

opportunity to test the precision of the 10Be ties. It may be worth including this comparison as a third panel to Fig. 3.

**The answer to this apparent paradox is very simple. Although the reviewer apparently missed it, our 10Be depths are indeed those of EDC96, as indicated in Table 1 (although we mistakenly gave as EDC 97). This is because both the volcanic and 10Be measurements were made in the core labeled as EDC 96. We show below the same plot as the referee, but using EDC 96 depths for both 10Be and the volcanic peaks of Severi et al. (2007). As can be seen, the agreement is excellent, even less than the 20 years we cite as our estimated precision. There in fact is some disagreement between authors of the present paper whether or not this is a true quantitative test of the uncertainty in our synchronization, as suggested by the reviewer. One of us (JJ) agrees with this argument, and in fact made it many years ago. Another (GMR) argues that in fact it only rigorously proves that our synchronization procedure independently aligns any chosen feature in NGRIP with a common 10Be peak in EDC and EDML (we must remember that EDML was dated using common volcanic peaks found in EDC, and thus any feature found in one must almost by construction be found at the correct depth in the other). While it highly likely that the 10Be peaks chosen in NGRIP correspond to those found in the Antarctic cores, there is no independent proof that this is the case. For example, let us consider a case where an anomalous peak in NGRIP is synchronized with a real 10Be peak in EDC. It will then, for the reason given above, also synchronize with high precision the same 10Be peak in EDML. While, as stated above, this example is unlikely, a more plausible possibility is that for some reason (higher accumulation, higher resolution sampling) the form of the 10Be peaks is different in NGRIP than in EDC/EDML. In that case, it is possible that our synchronization protocol will align a different part of the peak at NGRIP with that in EDC/EDML. If the 10Be peaks are due to solar minima, such as the Maunder Minimum, as we believe, they have durations of the order of 100 years. Thus, the above effect could lead to an offset of several decades between NGRIP and EDC/EDML, while still maintaining a tight correlation between EDC and EDML. In fact, this may be at least part of the explanation for one of the observations discussed in the paper, which is that there appears to be an offset corresponding to 27+/- 7 years between the observed depth in EDC/EDML for the NGRIP volcanic peaks L2 and L3 of Svensson et al. (2013), compared to the predicted depth using the 10Be synchronization.**

(4) In section 5 I am a confused by the different trends in the Δage and Δdepth. In my mind the two are exchangeable, as you can calculate one from the other using the ice chronology. For example, how is it possible that the EDC Δage for AICC2012 and Scen4 (red and black) are identical, while their Δdepth is so different? Doesn't that imply that these two chronologies have completely different annual layer thickness (while both are synched via 10Be)? The authors could provide a few more details on how the Δage and Δdepth are constructed, which may help in understanding what's going on. For example, which chronology is used for the 2 Loulergue scenarios? I would assume the authors ran the densification models using the AICC2012 T, Acc and chronology for consistency?

**The EDC3 (scenario 1 and 4) and AICC2012 Δage and Δdepth were directly taken from the official chronologies given respectively by Loulergue et al. (2007) for EDC3 (data available as supplementary material of this paper) and Bazin et al. (2013) and Veres et al. (2013) for AICC2012 (data available as supplementary material of this paper).**

**For each tie points on the depth scale, we took the Δage from the corresponding depth level directly from the published chronologies. Then, using the ice age corresponding to the depth level d1, we look at the depth d2 when the gas age at d2 equals the ice age at d1 on each chronology.**

**In the revised version, we will add some sentences to explain these figures.**

(5) In section 6 (P7 L25-27) the authors use the BREAKFIT routine and a MATLAB routine to estimate the breakpoint in the data. I don't think these routines are particularly fit for the problem at hand, given that the time series are short and very noisy. The data range must be picked to isolate AIM 10, and then the routines require the user to specify a range where they believe the breakpoint is located. Because it is short and noisy, these subjective choices seem to matter a lot for AIM 10. For example, I tried the fitting routine for just AIM 10 at WAIS (where I did this before), and got a timing of -10 or +205 years depending on whether I used linear or 2nd order fitting. I don't mean to suggest that the authors applied the code incorrectly, I simply want to highlight that for this particular problem the outcome is very sensitive to the subjective choices of the operator. The authors may have had the same experience.

For longer time series with less noise the routines perform well, and become independent of the subjective choices of the user.

The isotopic maxima that the authors identify in Fig. 7 can also be picked out by eye, so I suggest the authors just remove the fitting routines from the paper (my preference) or provide more details on how the fitting routines were applied (data range, etc) and how the uncertainties were estimated.

**Indeed, we have to specify a range for the detection of the breakpoint in the breakfit and matlab routines. In the previous manuscript, the chosen approach with the breakfit software was hence to have a ~ 500 year window moving between 42200 ka and 40500 ka and we took the first significant breakpoint. With the matlab routine, we chose first a 20-year window moving between +200 years and -200 years around the DO10 Greenland warming to determine the first breakpoint value. Then a 2-year window moving between ±100 years around this first breakpoint is applied for the 2ⁿᵈ order polynomial curve fitting. The breakpoint value for the WDC isotopic curve is quite sensitive to this second step and we find a value between 150 and 210 years before the mid-slope of the warming in NGRIP (a value of 250 years can be obtained but the fit is not coherent). It is indeed correct that a determination can also be made by eye and actually the breakfit and matlab routines are used to check if the breakpoint identified by eye is indeed statistically significant. Our aim in using such an approach was also to follow the same approach as in the WAIS community paper (2015).**

Minor/language:

Throughout the text: I would suggest replacing "delta age" and "delta depth" with Δage and Δdepth (i.e.using Greek Delta symbol) to confirm with common usage in the ice core literature. **Will do**

Throughout: WAIS Divide is spelled without a hyphen between "WAIS" and "Divide"
**Will do**
Title: "41 k" should probably be changed to "41 kyr/ka". Also, please include a hyphen in "beryllium-10". **Will do**

P1 L9: are these 2sigma uncertainty values?

**This estimate is a compromise between (1) the standard deviation (4+/-3 years) between EDC and EDML based on the independent 10Be synchronization with NGRIP, compared to the direct synchronization of Severi et al. (2007) as shown in the revised Fig shown below and (2) that (27+/-7 years) observed between the observed position of the presumed bipolar volcanic peaks L2 and L3 of Svensson et al. (2013) in EDC and EDML compared to their predicted position using the 10Be synchronization. As such it really does not have any 1 or 2 sigma meaning.**

P1 L 20: Remove "our". The author lists of Raisbeck et al. 2007 and Raisbeck et al. 2016 are not identical. **Will do**

P1 L 21: ... estimates of the DEPTH difference between ... **Will do**
P1 L25: In a previous study, Raisbeck et al. (2007) have .... (same reason as above) **Will do**

Section 3: Please specify confidence intervals for the uncertainty estimates. Are these 2 sigma?
**Same answer as above**

Section 3: Maybe note that the precision on the WAIS Divide CH4 interpolar synchronization at 40ka is estimated to be +/- 73 years (2 sigma uncertainty in Δage; see Buizert et al. 2015 Fig. 3e), and therefore the new 10Be synchro is more precise. **Will do**

P5 L12-14: Do you mean meteorology interferes with the actual atmospheric 10Be production rate, **No** or do you mean it leads to differences in deposition, transport and dilution? **Yes** Because of the annual layer count, Greenland Acc can be reconstructed much more accurately that Antarctic Acc – could this be one of the reasons the 200yr peak is better resolved at NGRIP?

**Possibly, but not obvious, since counted layers must be translated into surface accumulation using a thinning model, which at this depth involves multiplying by about a factor of 5, and might depend on temperature, for example between stadial and interstadial.**

P7 L21-22: I think it would be good to cite e.g. Blunier and Brook 2001 here, who were among the first to describe the asynchronous coupling clearly. **Will do**

Figs 7 and 8, caption: please specify how the WD2014 chronology was transferred to GICC05. I assume you divided by 1.0063 and then added 50 years to get from BP1950 to B2k?

**Yes, we follow this conversion described in Buizert et al. (2015) as cited in the captions of Figs 7 and 8.**

---

## Author Comment (AC2) · 26 Oct 2016

**Replies to Reviewer #2**

Raisbeck et al. present new high-resolution 10Be data (NGRIP, EDML & Vostok ice cores) that they use to synchronise ice core records from Greenland and Antarctica over the period of the Laschamp geomagnetic field minimum. They discuss the precision of this synchronisation and the phasing of water-isotope variations around D/O 10. In addition, they investigate cyclicities in the 10Be records and make an assessment of the ice age - gas age difference around this period.

I think this is a very interesting paper that shows new important 10Be records. I certainly recommend the publication of the manuscript in Climate of the Past after the authors addressed several important remarks that I have listed below. I also think that the publication of the paper should be linked to the requirement of publishing the 10Bedata as a supplementary table so that the analyses can be repeated with different methods.

**Yes, the 10Be data will be submitted as supplementary table.**

Comments:

Regarding the flux calculation there are three important comments that need to be addressed: 1: On Page 3, line 10 the authors say that the NGRIP accumulation rate is based on the ss09sea time scale. However, figure caption 1 says that the accumulation rate is based on the respective time scales (GICC05 for NGRIP). So there is a contradiction that needs to be sorted out. I think GICC05 should be used as time scale and as the basis for the accumulation rate.

**The reviewer is right. In fact, the flux shown as a histogram for NGRIP, and later smoothed, is indeed based on the GICC05 layer counting, corrected to surface accumulation using the ss06 thinning model. The reference to ss09sea is a "leftover" from a much earlier version of the paper. This will be clarified in the revised version.**

2: I do not understand that the flux calculations are based on a smoothed version of the accumulation rates (see page 3, line 8). This can introduce jumps in the 10Be flux at transitions (e.g. 10Be concentrations react immediately while a smoothed accumulation record follows slowly leading to artificial jumps in the fluxes). The flux calculation needs to be done on the raw data. Smoothing can then be done afterwards on the flux record.

**Whether flux should be calculated with raw or smoothed accumulation rates depends on how accurate the estimated accumulation rates are, and how rapidly they vary compared to the 10Be production. If they are exactly accurate, the reviewer is right that the raw accumulation values should be used. If they have large uncertainties compared to the production variations, use of raw values can in fact lead to artificial variations in flux. In practice, since we later smooth the flux, the choice is not critical for the present application, as discussed below. For NGRIP, where it is observed that temperatures and accumulation can vary in a decade or less, we have, as indicated above, indeed used the raw 11 cm layer counted values, even though we believe they introduce unrealistic variations. This is because the 11 cm samples in our record can represent as little as 4 years. Thus a missed or added layer will give a 25% error in accumulation, which thus accounts for a significant fraction of the high frequency variation in the histogram of Fig 2. As a test, we have thus tried smoothing the accumulation, but found that this does not lead to any significant difference in the smoothed flux shown in Fig 2. For EDC and**

**EDML, accumulation rates are based on models using correlation between accumulation and water isotopes. As can be seen in Fig 1, there is considerable high frequency variation in these water isotope values, which again we believe are not representative of actual accumulation variations. In addition to analytical error, they are probably due to such things as uneven seasonal variations, mixing of surface snow etc. We have thus decided to use the bag accumulation rates given in the references cited, which represents an effective smoothing of ~50 years. Once again we have tested different smoothing procedures, and found no significant difference in the smoothed fluxes. For Vostok, where we do not have water bag values for water isotopes, but do have delta D values for the same 10 cm samples used for 10Be, we use these and a correlation between delta D and accumulation obtained in other Vostok cores.**

3: It is not a priori clear that the 10Be flux indeed reflects the 10Be production rate only. It could have imprints of climate change (e.g. during D/O transitions). Therefore, the authors should discuss if the peaks they used for the synchronisation are robust, i.e. independent of flux calculation (also visible in the 10Be concentrations).

**While we believe there is considerable evidence that flux is a better measure of 10Be (and other trace species) deposition in polar cores, at the suggestion of the reviewer we have repeated the Match synchronizations of EDC and EDML with NGRIP using concentrations. We found that the peaks and valley corresponding to the tie points shown in Fig 2 did not vary by more than 20 years compared to those found using fluxes.**

**In summary we believe our 10Be synchronization is quite robust to the choice of raw or smoothed accumulations, or even the use of concentration.**

I do not want to question the synchronisation results. However, they seem to critically depend on the choice of the fix points. As visible in figure 2 there might be alternative choices for fix points due to the cyclic behaviour of the 10Be data. I would like that the authors discuss in more detail how they have chosen the fix points, how robust these choices are and if it could be possible that other choices could be done. If there are alternative choices it could influence the subsequent discussions.

**The reviewer is correct that synchronization using AnalySeries requires choosing tie points, which are necessarily subjective. This is why, as described in the text, we chose also to use where possible the MATCH protocol, which does not require choosing tie points.**

As an example, I would like that the authors to discuss the 3 youngest peaks in the Vostok data. The youngest peak is clearly not there. Could it be possible to shift the whole record 200 years younger which might lead to a better agreement of the peaks in the younger part of the record. This could be discussed in connection with the MATCH routine i.e. what are the reasons to exclude such scenarios. The authors mention the missing data in the Vostok record. However, for the periods of disagreement missing data does not seem to be the major problem.

**As discussed in the paper, the Vostok core could be synchronized only with AnalySeries (the MATCH protocol is more sensitive to gaps in the records being synchronized). Thus, other choices than the one shown could be imagined. Interestingly enough, at one point we actually tried shifting with AnalySeries the younger 3 peaks mentioned by the reviewer. However, we found that this implied a large and rapid change in accumulation**

**rate, which was not coincident with any stable isotope variation, and did not seem to us to be reasonable. Thus, while we cannot definitely exclude such a choice, we believe that it is less probable.**

I think the errors for the synchronisation are not well defined.

If I understand correctly they are based on the agreement of the 2 synchronisation methods. However, outliers can shift peaks and this could lead to biases in all synchronisation methods i.e. leading to a false sense of confirmation. I recommend that the author treat synchronisation errors with caution. An indication for systematic problems might be the systematic offset of 10Be-synchronised time scales and the identification of common volcanic induced spikes.

It is never discussed how the synchronisation errors should be understood. Does an error of <20 year correspond to a 1 sigma error or a 2 sigma error?

**No the errors are not based on the agreement between the two synchronization methods. This estimate is a compromise between (1) the standard deviation (4+/-3 years) between EDC and EDML based on the independent 10Be synchronization with NGRIP, compared to the direct synchronization of Severi et al. (2007) as shown in the revised Fig of the reply to Christo Buizert, and (2) that (27+/-7 years) observed between the observed position of the presumed bipolar volcanic peaks L2 and L3 of Svensson et al. (2013) in EDC and EDML compared to their predicted position using the 10Be synchronization. As such it really does not have any 1 or 2 sigma meaning.**

I think the spectral analysis is interesting but it shows that the results of Fourier analysis (especially significance levels) depend very sensitively on noise estimates (it is interesting that the noise levels are very different in the different records leading to the different results on the significance of cycles).

**We agree that the form of the noise levels, on which the significance levels depend, are very different for NGRIP compared to the Antarctic cores, possibly because of higher sample resolution.**

The wavelet analysis and the agreement of the different records indicate that there are common cycles irrespective of the significance analyses (otherwise the synchronisation would not work. . .).

**As discussed in the paper, we too were surprised at this apparent paradox, but can only note that the Antarctic cores seem to have longer period variations which distort the spectra.**

I think the authors should not attribute possible uncertainties to meteoric influences on 10Be records from Antarctica only (page 5 line 11), or do the authors want to imply that the Greenland data does not contain any meteorological influences?

**No, but they may be smaller because of a higher accumulation rate, or different deposition mechanism (higher fraction of wet deposition?). We will specify this.**

I have question regarding the approach for the methane synchronisation and delta age calculation. The authors synchronise the whole section (figure 5) but not single peaks. However, looking at methane from NGRIP one gets the impression that the data in the youngest period is placed rather too young. Can this explain the delta age difference between

the 10Be and other methods shown in figure 6.

**As the reviewer correctly notes, the methane peaks were matched over their whole profile using the Match protocol, which does not involve any subjective choices. Whether this accounts for the difference with the other methods is not obvious, but it is clear from Fig 5 that the relatively poor resolution of the methane data, particularly for EDML is the limiting factor in calculating delta age. Thus, although the present 10Be results have improved the ice synchronization by almost an order of magnitude, this has made only a modest improvement on the delta age uncertainty.**

I appreciate figure 7 since I think it shows the data in a honest way. It also shows that in my opinion, the results from the applied statistical tools needs to be treated with caution. For example, EDC shows a similar early isotope spike as WDC and therefore it appears to me that the lead/lag discussion is very much influenced by the noise in the data and the applied statistical tools. These uncertainties could be emphasised even more.

**We agree. We in fact wanted to emphasize that, looking at AIM-10 as synchronized with 10Be, and using the same statistical procedure as WAIS-Divide Project Members (2015), individual records give conflicting results.**

Regarding figure 8 and the discussion on longer time scales. I understand that this discussion fits into the discussion following figure 7. However, to me it feels like a step backwards. The authors have this great 10Be data and the synchronisation and then they go back to the results from an older synchronisation. Maybe figure 8 would fit better as figure 1 in connection to the introduction of the general topic.

**We accept the reviewer's suggestion to move Fig 8 earlier in the paper, which will of course require some modifications in the text.**

It is not 100% clear to me on what the errors in table 2 are based on. Uncertainties in the time scale synchronisation and/or in the method to find the transition points?

**The uncertainties are in the difference in the inflection points with respect to warming in Greenland, found using different parameters for BREAKFIT.**

Details:

I think the introduction is quite short but OK. I was also wondering why the authors list the results in the introduction.
**We wanted to give the reader enough information to decide whether it justifies his or her continuing reading (the "newspaper" rule).**

Page 2 line 19: Did the Vostok samples really weigh >500g? It seems like quite a lot even considering that the measurements were done a while ago.
**Yes. In fact the sampling was done more than 20 years ago.**

page 7 line 12: I would say "10Be-synchronised climate records"
**Will do**

In general, I think the paper is very well written!
**The comment is appreciated.**

---

## Author Response (AR1)

We are submitting here a revised version of the manuscript « An improved North-South synchronization of ice core records around the 41 kyr beryllium-10 peak**".** The point by point replies to the comments of the referees that were given in our previous responses have been implemented and are highlighted in the revised manuscript. We are also uploading a separate file with the 10Be data to be included as Supplementary Material, as requested.